

# Effects of age on the identification of emotions in facial expressions: a meta-analysis

Ana R. Gonçalves[1], Carina Fernandes[1,2,3], Rita Pasion[1], Fernando Ferreira-Santos[1], Fernando Barbosa[1] and João Marques-Teixeira[1]

[1] Laboratory of Neuropsychophysiology, Faculty of Psychology and Education Sciences, Universidade do Porto, Porto, Portugal
[2] Faculty of Medicine, Universidade do Porto, Porto, Portugal
[3] Language Research Laboratory, Institute of Molecular Medicine, Faculty of Medicine, Universidade de Lisboa, Lisboa, Portugal

## ABSTRACT

**Background**. Emotion identification is a fundamental component of social cognition. Although it is well established that a general cognitive decline occurs with advancing age, the effects of age on emotion identification is still unclear. A meta-analysis by Ruffman and colleagues (*2008*) explored this issue, but much research has been published since then, reporting inconsistent findings.

**Methods**. To examine age differences in the identification of facial expressions of emotion, we conducted a meta-analysis of 24 empirical studies ($N = 1,033$ older adults, $N = 1,135$ younger adults) published after 2008. Additionally, a meta-regression analysis was conducted to identify potential moderators.

**Results**. Results show that older adults less accurately identify facial expressions of anger, sadness, fear, surprise, and happiness compared to younger adults, strengthening the results obtained by *Ruffman et al. (2008)*. However, meta-regression analyses indicate that effect sizes are moderated by sample characteristics and stimulus features. Importantly, the estimated effect size for the identification of fear and disgust increased for larger differences in the number of years of formal education between the two groups.

**Discussion**. We discuss several factors that might explain the age-related differences in emotion identification and suggest how brain changes may account for the observed pattern. Furthermore, moderator effects are interpreted and discussed.

## INTRODUCTION

Emotion identification is defined as the "ability to visually analyze the configuration of facial muscle orientations and movements in order to identify the emotion to which a particular expression is most similar" (*Wilhelm et al., 2014*, p. 3) and is a central component of nonverbal communication. The ability to accurately identify emotional expressions is essential for successful interpersonal functioning throughout the lifespan (*Carstensen, Gross & Fung, 1997*). The interpretation of the emotions that others are experiencing is

Corresponding author
Ana R. Gonçalves,
a.s.ribeiro.g@gmail.com,
anasgoncalves@fpce.up.pt

important to avoid conflict and provide social support. Emotion identification ability is also fundamental to regulate behavior such as selectively attending and approaching to positively stimuli to elicit positive feelings and avoid negative ones (*Gross, Richards & John, 2006*). Importantly, presenting facial emotional stimuli is a valid and reliable approach in order to activate brain areas crucial for emotion processing (*Fusar-Poli et al., 2009*) and emotion identification tasks have been used in studies assessing emotional processing (*Ebner & Johnson, 2009*; *Gonçalves et al., 2018*; *Grady et al., 2007*; *Mienaltowski et al., 2011*; *Williams et al., 2006*).

A substantial body of research proposes an age-related "positivity effect" (*Mather & Carstensen, 2005*), defined as a tendency for older adults to attend to, and better memorize positive information relative to neutral and negative stimuli. According to the Socio-emotional Selectivity Theory (*Carstensen, Isaacowitz & Charles, 1999*), significant developmental changes occur in older adults' regulation and processing of affect. In this sense, the theory attributes the "positivity effect" to a motivational shift toward emotional regulation goals (i.e., achieving positive affect) as older adults begin to view their lifetime as limited (*Carstensen, Isaacowitz & Charles, 1999*). An alternative theoretical account of the age-related positivity effect, the dynamic integration theory, posits that greater cognitive demands required to process negative information lead older adults to automatically and preferentially process positive information (*Labouvie-Vief, 2003*).

A vast set of the literature shows emotion identification deficits in older adults (e.g., *Isaacowitz et al., 2007*; *Sullivan & Ruffman, 2004*). Furthermore, Ruffman and colleagues (*2008*) performed a meta-analysis to examine age differences in emotion identification across four modalities—faces, voices, bodies/contexts, and matching of faces to voices. Specifically in faces modality, Ruffman and colleagues (*2008*) found an age-related decline across all emotions, except for disgust. However, the mean effect sizes in the faces modality range from 0.07 to 0.34 across all emotions, reflecting inconsistencies among findings in the studies included. Following studies (*García-Rodríguez et al., 2009a*; *García-Rodríguez et al., 2009b*; *Orgeta, 2010*; *Suzuki & Akiyama, 2013*) also reported inconsistent findings, showing an age-related decline only in the identification of anger and fear (*García-Rodríguez et al., 2009a*; *García-Rodríguez et al., 2009b*) and anger and sadness (*Orgeta, 2010*), that raise again questions about the effects of age on emotion identification.

Human aging is accompanied by the decline of various cognitive abilities (for a review, see *Salthouse, 2009*). For example, sustained attention and working memory decrease with age (*Gazzaley et al., 2007*; *Park et al., 1996*). Importantly, these cognitive abilities seem to be relevant to the performance in emotion identification tasks (*Lambrecht, Kreifelts & Wildgruber, 2012*). Furthermore, aging has been linked to a gradual reduction in visual acuity (*Caban et al., 2005*; *Humes et al., 2009*). Despite the well-known age-related decline in certain cognitive and sensory functions and its possible influence on emotion identification, the effects of age on emotion identification abilities remain unclear.

Analyzing studies published after 2008, the present meta-analysis aims to clarify whether age-related difficulties in identifying facial emotional expressions exist, quantify the magnitude of age effects observed and identify potential moderators.
There are several factors known to influence the identification of facial expressions. Specifically, studies focusing on emotional facial expressions support the idea of a female advantage in emotion identification (*Hall & Matsumoto, 2004*; *Montagne et al., 2005*; *Williams et al., 2009*). Furthermore, participants with no college education ($M_{age} = 35.5$, $SD = 13.1$, range $= 19$–$69$ years) were more likely to select the correct label for anger and sadness, than were those with a college degree ($M_{age} = 33.9$, $SD = 11.0$, range $= 19$–$64$ years). For fear and disgust, the opposite pattern was reported (*Trauffer, Widen & Russell, 2013*). Besides participants characteristics, stimulus features need to be considered when analyzing different studies of emotion perception. For instance, color has been reported to improve the perception of general emotional clues (*Silver & Bilker, 2015*). Additionally, dynamic stimuli can be more accurately recognized than the static ones as shown by behavioral studies (*Ambadar, Schooler & Cohn, 2005*). Considering that most real-word emotion recognition involves motion of the perceiver and the target rather than looking at pictures, using dynamic stimuli in research makes sense (*Isaacowitz & Stanley, 2011*). Another element that may contribute to the differential interpretation of static and dynamic facial expressions is motivation, particularly in older adults, since a static photo may create a perception of an overly artificial task, as well as very different from daily life, so that older adults may not engage sufficiently to perform well (*Isaacowitz & Stanley, 2011*). Given these evidences, the variables sex, level of education of participants, and stimulus features (virtual vs natural, color vs black and white, static vs dynamic) were tested as moderators of any age effects observed. We expected to find larger effects for larger differences in the mean years of education between the groups to be compared, as well as for higher percentage of female participants and dynamic colored pictures of faces. With the present study, we will clarify how emotion identification of facial expressions changes along aging and identify potential moderators.

## MATERIALS & METHODS

### Literature search

A computer-based search of the PubMed, Web of Knowledge, and EBSCOhost (including the Academic Search Complete, PsycARTICLES, Psychology and Behavioral Sciences databases) was conducted in October 2017 by two researchers (ARG, CF). The search expression was "(aging OR ageing OR "older adults" OR elderly) AND ("emotion recognition*" OR "emotional processing" OR "emotion identification")". The search was limited to titles and abstracts, published in English in the last nine years. In PubMed the filter "Humans" was also used. A total of 1580 non-duplicated articles were found. Additionally, the references of the included articles were searched manually to identify other relevant studies ($n = 20$).

### Selection criteria

Studies assessing emotion identification in healthy younger ($20 \leq$ mean age $\leq 35$) and older adults (mean age $\geq 55$ years old) were included (criterion 1). Also, only studies that allowed effect size data (i.e., sample sizes, means, and standard deviations) to be directly recorded, calculated, or measured (i.e., from a graph) were included. Authors were
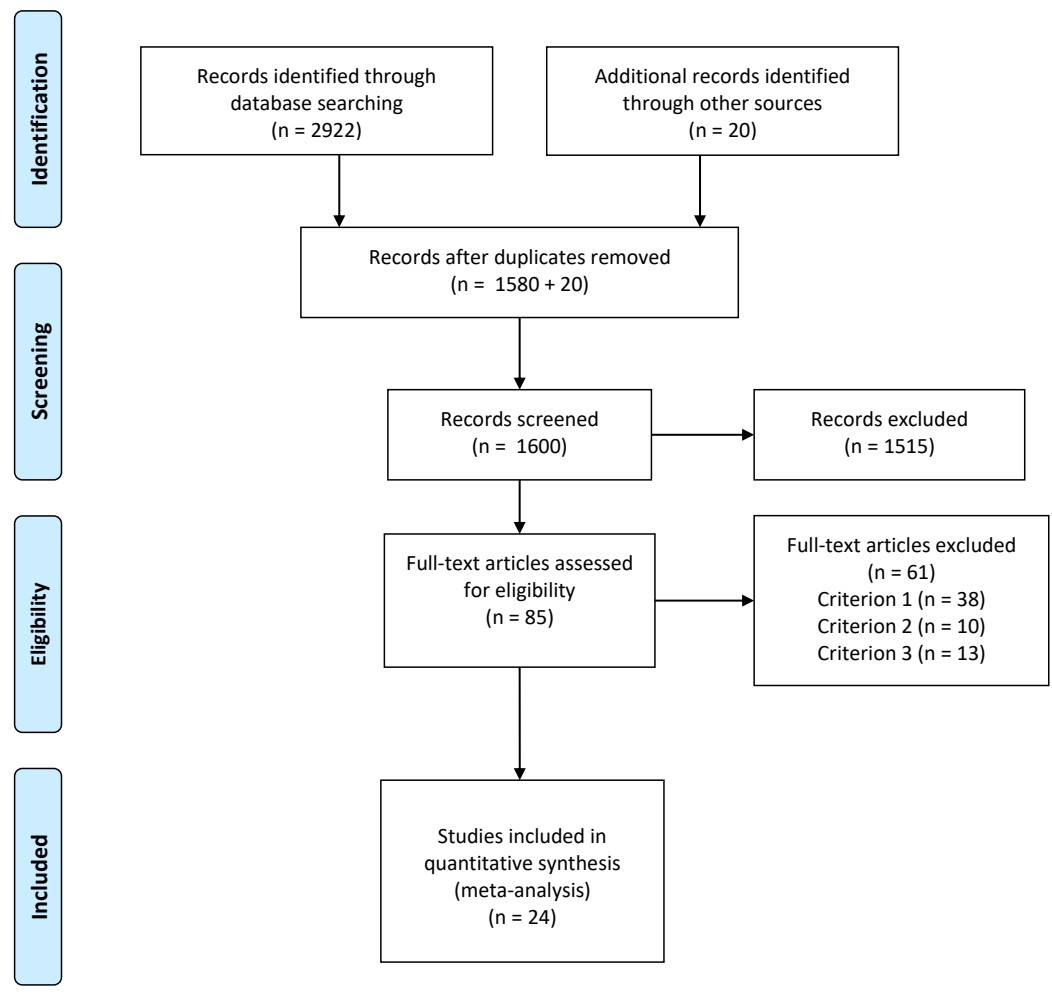

**Figure 1  PRISMA flow diagram.**

contacted if effect sizes could not be obtained from the published data. Ten studies that did not present descriptive statistics and the information requested was not provided, were excluded (criterion 2). Studies that did not guarantee the neurological and psychological health of the participants, or had missing details about the participants' inclusion criteria, were excluded ($n = 13$; criterion 3).

After screening for relevant studies ($n = 1,600$), considering the title and abstract, two researchers (ARG, CF) read the full-text of the studies that were retained ($n = 85$) and, independently, decided their eligibility for further analysis. Disagreements were resolved by consensus. The inter-rater agreement Cohen's kappa was used to compare agreement between the researchers, revealing an almost perfect agreement ($k = .95$).

Detailed information on the study selection process is described in the PRISMA Flow Diagram (Fig. 1).

## Recorded variables and data collection

The data of each paper were added to an extraction sheet, developed for this meta-analysis and refined when necessary.

When present, the following variables were extracted from each paper: (a) characteristics of the sample (sample groups, sample size, number of female participants, age, years of education); (b) emotion identification tasks and conditions; (c) descriptive statistics of participants' performance; (d) significant statistical differences between younger and older adults' performance.

## Statistical analysis

The Standard Mean Difference (*SMD*), based on Hedges' adjusted *g* formulation, was used to assess the association between the two variables of interest, i.e., how much age-groups' performance differ on the emotion identification task. The *SMD* was pooled across studies to derive an estimate of the mean (i.e., effect size based on Hedges' *g*), with each effect weighted for precision to correct for sampling error. To do so, a random-effects model was adopted.

Heterogeneity across the studies was tested using the $I^2$ and $Q$ statistics. Methodological and sample characteristics of the studies included in the meta-analysis are detailed in Table 1. Publication bias was assessed by visual inspection of the funnel plot. Egger's tests were used to estimate the severity of publication bias, with $p < .05$ considered statistically significant.

For each emotional expression, the unrestricted maximum likelihood random-effects meta-regression of the effect size was performed with sex (% female), differences in the level of education between older and younger adults, and stimulus features (virtual vs natural, color vs black and white, static vs dynamic) as moderators to determine whether these covariates influenced the effect size.

Statistical analyses were performed using Cochrane Collaboration Review Manager 5.3 (The Nordic Cochrane Centre, The Cochrane Collaboration, 2014) and SPSS version 22.0 (IBM Corp, 2013) software.

## RESULTS

The negative overall effect size for *age-group* across all emotions ($M = -1.80$) showed that facial expressions were less accurately identified by older adults (Table 2). For each effect size, a negative value indicates that older adults have performed worse than younger adults, whereas a positive value indicates the reverse. When analyzing data by emotion, the combined effect sizes showed that facial expressions of anger, sadness, fear, surprise, and happiness were less accurately identified by older adults (Table 2). Regarding the identification of facial expressions of disgust, no significant differences were found between older and younger adults (Table 2).

Significant heterogeneity was found for all emotions, indicating that the effects contributing to each of the estimates differ substantively. Effect sizes for individual studies are depicted in Table 3.

**Table 1  Methodological and sample characteristics of studies included in the meta-analysis.**

| Study | Condition | Paradigm | Stimuli | Sample size Older (F) | Younger (F) | Mean age Older | Younger | Mean years of education Older | Younger |
|---|---|---|---|---|---|---|---|---|---|
| *Baena et al. (2010)* | Human faces | EIT | VHF | 39 | 39 | 69.9 | 23.7 | | |
| *Campbell et al. (2015)* | Directed gaze/YF | EIT | FACC | 32(15) | 32(15) | 71.0 | 20.4 | | |
| *Carvalho et al. (2014)* | | EIT | FACC | 17(12) | 12(6) | 73.6 | 34.5 | 7.43 | 8.34 |
| *Chaby et al. (2015)* | | EIT | FACC | 31(17) | 31(16) | 67.2 | 25.8 | 13.6 | 14.2 |
| *Circelli, Clark & Cronin-Golomb (2013)*/ Study 1 | | EIT | GFAC | 16(9) | 16(11) | 68.9 | 19.2 | 16.4 | 13.9 |
| *Ebner, He & Johnson (2011)* | YF | EIT | FACC | 51(24) | 52(27) | 73.6 | 26.0 | | |
| *Ebner, Johnson & Fischer (2012)* | YF | EIT | FACC | 30(17) | 30(16) | 68.2 | 25.1 | 14.5 | 14.8 |
| *García-Rodríguez et al. (2009a)* and *García-Rodríguez et al. (2009b)* | | EIT | FACC | 16(8) | 16(8) | 73.2 | 28.0 | 8.00 | 15.7 |
| *Halberstadt et al. (2011)* | Faces | EIT | GFAC | 61(36) | 60(34) | 70.5 | 20.5 | | |
| *Hunter, Phillips & Macpherson (2010)*/ Study 1 | | EIT | VHF | 25(10) | 25(16) | 67.0 | 22.6 | 15.4 | 15.1 |
| *Krendl & Ambady (2010)*/ Study 1 | | DANVA2 | FACC | 42(29) | 36(21) | 75.8 | 19.8 | | |
| *Krendl, Ambady & Rule (2014)* | | DANVA2M | FACC | 30(21) | 32(26) | 70.7 | 23.1 | 16.7 | 16.2 |
| *Lambrecht, Kreifelts & Wildgruber (2012)* | Visual stimulus | EIT | VFAC | 17(8) | 16(8) | | | | |
| *Murphy & Isaacowitz (2010)* | | DANVA2 | FACC | 23(15) | 41(22) | 72.0 | 19.3 | | |
| *Ngo & Isaacowitz (2015)*/Study 1 | Neutral context | EIT | FACC | 30(19) | 31(19) | | | | |
| *Noh & Isaacowitz (2013)* | Neutral context | EIT | GFAC | 47(39) | 37(23) | | | 18.1 | 15.7 |
| *Orgeta (2010)* | | EIT | GFAC | 40(27) | 40(27) | 69.7 | 22.4 | 14.0 | 14.5 |
| *Sarabia-Cobo et al. (2015)* | EIP | EIT | VHF | 37(21) | 50(26) | 72.3 | 28.5 | 8.30 | 16.9 |
| *Silver & Bilker (2015)* | Visual stimulus | EIT | FACC | 39(0) | 37(0) | 72.8 | 33.5 | 15.8 | 11.0 |
| *Sullivan et al. (2015)* | | EIT | FACC | 58(30) | 60(30) | 70 | 20 | | |
| *Suzuki & Akiyama (2013)* | | EIT | GFAC | 36(18) | 36(18) | 69.4 | 21.4 | 14.2 | 14.4 |
| *Svärd, Wiens & Fischer (2012)* | | URT | GFAC | 20(10) | 19(10) | 73.7 | 26.4 | 14.2 | 14.4 |
| *Williams et al. (2009)* | | EEI | FACC | 276(140) | 176(111) | | | | |
| *Ziaei et al. (2016)* | Direct gaze | EIT | GFAC | 20(10) | 20(10) | 69.8 | 20.6 | 15.3 | 14.3 |

**Notes.**
Condition: YF, young faces; EIP, emotional intensity pronounced.
Paradigm: EIT, emotion identification task; DANVA2, DANVA2 adults face task; DANVA2 M, DANVA2 modified task; URT, unmasked recognition task; EEI, explicit emotion identification.
Stimuli: VHF, virtual human faces; FACC, colour photos of human faces; GFAC, grey scale photos of human faces; VFAC, video sequences of human faces.
Sample size: F, number of females.

Egger's regression tests showed no significant funnel plot asymmetry across emotional expressions, indicating the inexistence of publication bias.

The meta-regression analyses showed a significant association between participants' performance by *age-group* and both *sex* and *level of education* as moderators on fear and disgust identification (Table 4). Specifically, differences in *level of education* are associated with effect sizes on the identification of fear and disgust expressions, with larger effects observed for larger differences in education. Regarding the moderator *sex*, larger effects are observed for higher percentages of female participants on the identification of fear

**Table 2  Age effects for recognition of different emotions**

|  | M | K | N | $I^2$ |
|---|---|---|---|---|
| Anger | −0.61[***] | 21 | 1,785 | .76[***] |
| Sadness | −0.43[***] | 18 | 1,661 | .64[***] |
| Fear | −0.62[***] | 18 | 1,606 | .53[**] |
| Disgust | −0.04 | 16 | 1,480 | .88[***] |
| Surprise | −0.45[***] | 9 | 621 | .90[***] |
| Happiness | −0.19[*] | 22 | 1832 | .70[***] |
| Overall | −1.80[***] | 24 | 1978 | .98[***] |

Notes.

M, mean effect size; K, number of independent studies contributing towards each respective mean effect size. A negative effect size denotes that older adults are worse than younger adults; a positive effect size indicates the reverse. N, number of participants. $I^2$ quantifies within-group heterogeneity.

Significances are marked by $^*p < .05$, $^{**}p < .01$, and $^{***}p < .001$.

and the opposite pattern (i.e., larger effects are observed for smaller percentages of female participants) is observed on the identification of disgust expression. A significant association was also found between *stimulus features* (virtual vs natural, color vs black and white, static vs dynamic) as moderator and performance by *age-group* on disgust identification. Concerning fear identification the association was marginally significant (Table 4). Whereas larger effects are observed for grayscale pictures of faces on the identification of disgust, larger effects are observed for virtual faces on the identification of fear.

## DISCUSSION

The present study aimed to identify potential age-related differences in identifying emotions in facial expressions and quantify the magnitude of the observed age effects. Using a meta-analytic approach with a random-effect model, our results showed that older adults identified facial expressions of anger, sadness, fear, surprise, and happiness less accurately than younger adults. In contrast, identification of disgust appears to be preserved with age, as older and younger adults' performance was similar in this case. The present results support those reported in a prior meta-analysis by *Ruffman et al. (2008)*.

Taken together, our results are consistent with a general emotion identification decline associated with aging. Thus, this meta-analysis does not support a positivity bias in the identification of facial expressions of emotion, as impairments in this ability seem to extend to positive facial expressions, nor previous findings suggesting that aging is associated with a reduction in the negativity effect, rather than a positivity effect (*Comblain, D'Argembeau & Van der Linden, 2005*; *Denburg et al., 2003*; *Knight, Maines & Robinson, 2002*; *Mather et al., 2004*). Age-related positivity effects were found primarily in attention to, and recall and recognition memory for emotional images which could have implications for emotion identification (*Isaacowitz & Stanley, 2011*). Therefore, several studies aimed to investigate whether age differences in emotion identification performance could also reflect positivity effects (e.g., *Williams et al., 2006*). Importantly, many tasks assessing identification accuracy for positive emotions are constrained by ceiling effects (due to the relative low difficulty

**Table 3  Effect size data for individual studies included in the meta-analysis.**

| Study | Sample size | Weight (%) | Effect size [95% CI] |
|---|---|---|---|
| *Anger* | | | |
| *Campbell et al. (2015)* | 64 | 4.9 | −0.74 [−1.25, −0.23] |
| *Carvalho et al. (2014)* | 29 | 3.3 | −1.32 [−2.14, −0.49] |
| *Chaby et al. (2015)* | 62 | 4.8 | −0.87 [−1.40, −0.35] |
| *Circelli, Clark & Cronin-Golomb (2013)* | 32 | 3.9 | 0.30 [−0.40, 1.00] |
| *Ebner, Riediger & Lindenberger (2010)* | 103 | 5.4 | −0.87 [−1.27, −0.47] |
| *Ebner, Johnson & Fischer (2012)* | 60 | 4.9 | 0.15 [−0.36, 0.65] |
| *García-Rodríguez et al. (2009a)* and *García-Rodríguez et al. (2009b)* | 32 | 2.6 | −2.84 [−3.86, −1.83] |
| *Halberstadt et al. (2011)* | 121 | 5.7 | −0.51 [−0.88, −0.15] |
| *Hunter, Phillips & Macpherson (2010)*/Study 1 | 50 | 4.4 | −1.00 [−1.59, −0.41] |
| *Krendl & Ambady (2010)*/Study 1 | 78 | 5.2 | −0.63 [−1.09, −0.17] |
| *Krendl, Ambady & Rule (2014)* | 62 | 4.8 | −0.93 [−1.45, −0.40] |
| *Lambrecht, Kreifelts & Wildgruber (2012)* | 33 | 3.6 | −1.31 [−2.07, −0.55] |
| *Murphy & Isaacowitz (2010)* | 64 | 4.7 | −0.98 [−1.52, −0.44] |
| *Ngo & Isaacowitz (2015)*/Study 1 | 61 | 4.8 | −0.80 [−1.32, −0.27] |
| *Noh & Isaacowitz (2013)* | 84 | 5.3 | −0.41 [−0.85, 0.02] |
| *Orgeta (2010)* | 80 | 5.3 | 0.00 [−0.44, 0.44] |
| *Sarabia-Cobo et al. (2015)* | 87 | 5.3 | −0.51 [−0.95, −0.08] |
| *Sullivan et al. (2015)* | 118 | 5.7 | −0.28 [−0.64, 0.08] |
| *Suzuki & Akiyama (2013)* | 72 | 5.0 | −0.72 [−1.20, −0.24] |
| *Williams et al. (2009)* | 452 | 6.4 | −0.64 [−0.83, −0.44] |
| *Ziaei et al. (2016)* | 40 | 4.1 | 1.06 [0.39, 1.73] |
| *Sadness* | | | |
| *Baena et al. (2010)* | 78 | 5.9 | −0.09 [−0.53, 0.36] |
| *Campbell et al. (2015)* | 64 | 5.4 | −0.36 [−0.85, 0.14] |
| *Carvalho et al. (2014)* | 29 | 3.5 | 0.21 [−0.53, 0.95] |
| *Chaby et al. (2015)* | 62 | 5.1 | −1.02 [−1.55, −0.49] |
| *Circelli, Clark & Cronin-Golomb (2013)* | 32 | 3.8 | −0.34 [−1.04, 0.36] |
| *Ebner, Riediger & Lindenberger (2010)* | 103 | 6.5 | −0.37 [−0.76, 0.02] |
| *García-Rodríguez et al. (2009a)* and *García-Rodríguez et al. (2009b)* | 32 | 3.7 | 0.65 [−0.06, 1.37] |
| *Halberstadt et al. (2011)* | 121 | 6.8 | −0.29 [−0.65, 0.07] |
| *Hunter, Phillips & Macpherson (2010)*/Study 1 | 50 | 4.6 | −0.91 [−1.50, −0.33] |
| *Krendl & Ambady (2010)*/Study 1 | 78 | 5.8 | −0.48 [−0.93, −0.03] |
| *Krendl, Ambady & Rule (2014)* | 62 | 5.3 | −0.57 [−1.08, −0.06] |
| *Murphy & Isaacowitz (2010)* | 64 | 5.2 | −0.61 [−1.14, −0.09] |
| *Orgeta (2010)* | 80 | 5.5 | −1.38 [−1.87, −0.89] |
| *Sarabia-Cobo et al. (2015)* | 87 | 6.0 | −0.47 [−0.90, −0.04] |
| *Silver & Bilker (2015)* | 76 | 5.8 | −0.56 [−1.02, −0.10] |
| *Sullivan et al. (2015)* | 118 | 6.8 | −0.28 [−0.64, 0.08] |
| *Suzuki & Akiyama (2013)* | 72 | 5.7 | −0.54 [−1.01, −0.07] |
| *Williams et al. (2009)* | 452 | 8.6 | −0.13 [−0.32, 0.06] |

**Table 3** (*continued*)

| Study | Sample size | Weight (%) | Effect size [95% CI] |
|---|---|---|---|
| *Fear* | | | |
| *Campbell et al. (2015)* | 64 | 5.5 | −0.46 [−0.96, 0.04] |
| *Carvalho et al. (2014)* | 29 | 3.3 | −0.22 [−0.96, 0.52] |
| *Chaby et al. (2015)* | 62 | 5.4 | −0.36 [−0.86, 0.15] |
| *Circelli, Clark & Cronin-Golomb (2013)*/Study 1 | 32 | 3.6 | −0.52 [−1.23, 0.18] |
| *Ebner, Riediger & Lindenberger (2010)* | 103 | 6.9 | −0.50 [−0.89, −0.11] |
| *García-Rodríguez et al. (2009a)* and *García-Rodríguez et al. (2009b)* | 32 | 2.7 | −1.93 [−2.79, −1.07] |
| *Halberstadt et al. (2011)* | 121 | 7.4 | −0.07 [−0.43, 0.28] |
| *Hunter, Phillips & Macpherson (2010)*/Study 1 | 50 | 4.7 | −0.61 [−1.18, −0.04] |
| *Krendl & Ambady (2010)*/Study 1 | 78 | 6.0 | −0.60 [−1.06, −0.14] |
| *Krendl, Ambady & Rule (2014)* | 62 | 5.5 | −0.10 [−0.59, 0.40] |
| *Murphy & Isaacowitz (2010)* | 64 | 4.8 | −1.23 [−1.79, −0.68] |
| *Ngo & Isaacowitz (2015)*/Study 1 | 61 | 5.2 | −0.75 [−1.27, −0.23] |
| *Orgeta (2010)* | 80 | 6.1 | −0.52 [−0.97, 0.08] |
| *Sarabia-Cobo et al. (2015)* | 87 | 6.0 | −1.04 [−1.50, −0.59] |
| *Sullivan et al. (2015)* | 118 | 7.2 | −0.65 [−1.02, −0.28] |
| *Suzuki & Akiyama (2013)* | 72 | 5.6 | −0.98 [−1.48, −0.49] |
| *Svärd, Wiens & Fischer (2012)* | 39 | 4.0 | −0.66 [−1.31, −0.02] |
| *Williams et al. (2009)* | 452 | 10.0 | −0.64 [−0.83, −0.44] |
| *Disgust* | | | |
| *Campbell et al. (2015)* | 64 | 6.4 | −0.39 [−0.88, 0.11] |
| *Carvalho et al. (2014)* | 29 | 6.0 | −0.57 [−1.33, 0.18] |
| *Chaby et al. (2015)* | 62 | 5.5 | 0.92 [0.39, 1.44] |
| *Circelli, Clark & Cronin-Golomb (2013)*/Study 1 | 32 | 6.1 | 0.31 [−0.39, 1.01] |
| *Ebner, Riediger & Lindenberger (2010)* | 103 | 6.5 | −0.56 [−0.95, −0.16] |
| *García-Rodríguez et al. (2009a)* and *García-Rodríguez et al. (2009b)* | 32 | 5.9 | −0.38 [−1.08, 0.32] |
| *Halberstadt et al. (2011)* | 121 | 6.5 | 0.72 [0.36, 1.09] |
| *Hunter, Phillips & Macpherson (2010)*/Study 1 | 50 | 6.3 | −0.11 [−0.67, 0.44] |
| *Lambrecht, Kreifelts & Wildgruber (2012)* | 33 | 6.0 | −1.53 [−2.31, −0.74] |
| *Ngo & Isaacowitz (2015)*/Study 1 | 61 | 6.1 | −0.49 [−1.00, 0.02] |
| *Noh & Isaacowitz (2013)* | 84 | 6.3 | 0.31 [−0.12, 0.75] |
| *Orgeta (2010)* | 80 | 6.3 | 0.37 [−0.08, 0.81] |
| *Sarabia-Cobo et al. (2015)* | 87 | 6.4 | −1.21 [−1.67, −0.74] |
| *Sullivan et al. (2015)* | 118 | 6.5 | 0.30 [−0.06, 0.66] |
| *Suzuki & Akiyama (2013)* | 72 | 6.4 | 0.58 [0.11, 1.05] |
| *Williams et al. (2009)* | 452 | 6.6 | 0.62 [0.43, 0.81] |

**Table 3** (*continued*)

| Study | Sample size | Weight (%) | Effect size [95% CI] |
|---|---|---|---|
| *Surprise* | | | |
| *Carvalho et al. (2014)* | 29 | 4.8 | −0.62 [−1.38, 0.14] |
| *Circelli, Clark & Cronin-Golomb (2013)* | 32 | 5.7 | −0.02 [−0.72, 0.67] |
| *García-Rodríguez et al. (2009a)* and *García-Rodríguez et al. (2009b)* | 32 | 3.5 | 2.08 [1.20, 2.96] |
| *Halberstadt et al. (2011)* | 121 | 21.6 | 0.00 [−0.36, 0.36] |
| *Hunter, Phillips & Macpherson (2010)*/Study 1 | 50 | 8.4 | −0.67 [−1.24, −0.10] |
| *Orgeta (2010)* | 80 | 15.0 | −0.32 [−0.75, 0.11] |
| *Sarabia-Cobo et al. (2015)* | 87 | 9.2 | −2.06 [−2.61, −1.51] |
| *Sullivan et al. (2015)* | 118 | 20.7 | −0.39 [−0.76, −0.03] |
| *Suzuki & Akiyama (2013)* | 72 | 11.2 | −1.07 [−1.57, −0.58] |
| *Happiness* | | | |
| *Baena et al. (2010)* | 78 | 5.0 | 0.05 [−0.39, 0.49] |
| *Campbell et al. (2015)* | 64 | 4.7 | 0.00 [−0.49, 0.49] |
| *Carvalho et al. (2014)* | 29 | 3.3 | 0.17 [−0.57, 0.91] |
| *Chaby et al. (2015)* | 62 | 4.6 | −0.14 [−0.64, 0.36] |
| *Circelli, Clark & Cronin-Golomb (2013)* | 32 | 3.5 | 0.04 [−0.66, 0.73] |
| *Ebner, Riediger & Lindenberger (2010)* | 103 | 5.3 | −0.29 [−0.68, 0.10] |
| *Ebner, Johnson & Fischer (2012)* | 60 | 4.5 | 0.37 [−0.14, 0.88] |
| *García-Rodríguez et al. (2009a)* and *García-Rodríguez et al. (2009b)* | 32 | 3.5 | −0.39 [−1.09, 0.31] |
| *Halberstadt et al. (2011)* | 121 | 5.5 | −0.03 [−0.39, 0.33] |
| *Hunter, Phillips & Macpherson (2010)*/Study 1 | 50 | 4.2 | −0.42 [−0.98, 0.14] |
| *Krendl & Ambady (2010)*/Study 1 | 78 | 4.9 | −0.47 [−0.93, −0.02] |
| *Krendl, Ambady & Rule (2014)* | 62 | 4.6 | −0.40 [−0.91, 0.10] |
| *Lambrecht, Kreifelts & Wildgruber (2012)* | 33 | 2.8 | −1.95 [−2.79, −1.10] |
| *Murphy & Isaacowitz (2010)* | 64 | 4.5 | 0.07 [−0.44, 0.58] |
| *Orgeta (2010)* | 80 | 5.0 | 0.16 [−0.28, 0.60] |
| *Sarabia-Cobo et al. (2015)* | 87 | 5.1 | −0.44 [−0.87, −0.01] |
| *Silver & Bilker (2015)* | 76 | 4.9 | −0.24 [−0.69, 0.21] |
| *Sullivan et al. (2015)* | 118 | 5.5 | −0.38 [−0.75, −0.02] |
| *Suzuki & Akiyama (2013)* | 72 | 4.6 | 1.07 [0.58, 1.57] |
| *Svärd, Wiens & Fischer (2012)* | 39 | 3.8 | −0.61 [−1.26, 0.03] |
| *Williams et al. (2009)* | 452 | 6.5 | −0.03 [−0.22, 0.15] |
| *Ziaei et al. (2016)* | 40 | 3.6 | −1.21 [−1.89, −0.53] |

of the task); however, in the present data, the typical ceiling effects in younger adults' happiness recognition (e.g., *Williams et al., 2006*) seem to be absent.

Furthermore, our meta-regression results showed a significant association between sample characteristics, namely the proportion of female participants and the level of education, and participants' performance by age-group on the identification of fear and disgust. Stimulus features were also found to be significantly associated with participant's performance by age-group on disgust identification. Concerning fear identification, the association was marginally significant. Regarding the level of education, the effect size

**Table 4  Effect of moderators on the age-related differences in emotion recognition.**

|  | Q | df | p | Moderator | Z | p | β |
|---|---|---|---|---|---|---|---|
| *Anger* | | | | | | | |
| Model | 1.28 | 3 | .734 | | | | |
| *Sadness* | | | | | | | |
| Model | 3.09 | 3 | .377 | | | | |
| *Fear* | 34.0 | 3 | .000 | | | | |
| Model | | | | | | | |
| | | | | Sex (%F) | 2.06 | .039 | .35 |
| | | | | Mean Years of Educat. Dif. Stimulus | 4.12 | .000 | .78 |
| | | | | | −1.86 | .062 | −.32 |
| *Disgust* | | | | | | | |
| Model | 22.4 | 3 | .000 | | | | |
| | | | | Sex (%F) | −2.28 | .023 | −.52 |
| | | | | Mean Years of Educat. Dif. | 2.86 | .004 | .66 |
| | | | | Stimulus | 2.40 | .016 | .55 |
| *Surprise* | | | | | | | |
| Model | 1.25 | 3 | .742 | | | | |
| *Happiness* | | | | | | | |
| Model | 0.54 | 3 | .910 | | | | |

**Notes.**

Moderator: %F, percentage of female..

increases for larger differences in the mean years of education between the two groups. This result is consistent with the pattern reported by Trauffer and colleagues (*2013*) in which participants with college education were more likely to select the correct label for fear and disgust, than were those with no college degree. According to the authors (*Trauffer, Widen & Russell, 2013*), the number of correct and incorrect responses is partially influenced by the tendency to use certain labels. For instance, sadness and ager have a broader meaning for preschoolers than for university undergraduates which matches with the more frequent use of these words by participants with no college education, compared to the ones with a college education (*Trauffer, Widen & Russell, 2013*). With respect to the moderator sex, the pattern of effects observed suggests that female participants had better performance than male participants when identifying fear expression and worst performance when identifying disgust. For the identification of fear, the result is consistent with the idea of a female advantage in overall emotion identification supported by studies focusing on emotional facial expressions (*Hall & Matsumoto, 2004*; *Montagne et al., 2005*; *Williams et al., 2009*). For the identification of disgust, the result may be explained by the higher value of within-group heterogeneity found in the analysis of disgust expression ($I^2_{\mathrm{disgust}} = .880$ vs. $I^2_{\mathrm{fear}} = .053$). Contrary to what was expected, the meta-regression results of stimulus features suggest that disgust was better identified on grayscale pictures and fear was better identified on virtual faces. However, it should be noted that the report of color to improve the perception of emotional clues (*Silver & Bilker, 2015*) refers to general emotional clues and not to one specific emotion. The better identification of fear on virtual faces may be

explained by less variability in expressive features, compared to natural faces, which means by containing less noise (*Dyck et al., 2008*). Nevertheless, a note of caution should be added here. Results of regression-based methods may not be robust in the current meta-analysis, as such methods are more accurate with a larger number of studies.

Studies that explored the neural basis of emotion processing, either in younger or older adults, present evidence that brain changes might be responsible for alterations in emotion identification performance (*Brassen, Gamer & Büchel, 2011*; *Delgado et al., 2008*; *Ge et al., 2014*; *Murty et al., 2009*; *Urry et al., 2009*). In particular, the prefrontal cortex and amygdala were found to be key players in the neural mechanisms underlying emotional regulation (*Delgado et al., 2008*; *Murty et al., 2009*). Mather and colleagues (*2004*) reported reduced amygdala activation for pictures of negative valence during their encoding in older adults. The authors suggested that the on-line reductions in response to negative pictures should cause disproportionately reduced subsequent memory for these negative stimuli. This pattern of amygdala activation was also found by Keightley and colleagues (*2007*). Our results regarding the identification of negative expressions, except for the identification of disgust, are consistent with the abovementioned evidence. Besides a general reduction of the amygdala response, according to *Ruffman et al. (2008)*, the increased difficulty of older adults to recognize facial expressions of anger may be related to a functional decline in the orbitofrontal cortex, sadness to a decline in the cingulate cortex and amygdala, and fear to a decline in the amygdala. Nevertheless, the identification of neural circuits rather than specific brain regions might be more successful when trying to explain the differences found between younger and older adults' performance (*Almeida et al., 2016*; *Barrett & Wager, 2006*; *Clark-Polner, Johnson & Barrett, 2016*), including the identification of positive expressions.

Impairments in cognitive and sensory functions might also explain the changes in emotion identification across the lifespan. Aging is often accompanied by a decline in cognitive abilities (for review, see *Salthouse, 2009*), as well as by losses in visual and auditory acuity (*Caban et al., 2005*; *Humes et al., 2009*), which could hinder higher-level processes such as language and perception (*Sullivan & Ruffman, 2004*). However, these sensory features have been reported to be poor predictors of the decline in visual or auditory emotional identification that occurs with aging (e.g., *Lima et al., 2014*; *Ryan, Murray & Ruffman, 2010*). We could not examine these putative moderators due to a lack of consistent selection of cognitive ability measures and its reporting across studies. Future studies incorporating common measures of cognitive ability would allow addressing this issue.

As a final note, we highlight the ambiguity of emotion identification and emotion recognition concepts in the literature. Some studies used both terms interchangeably (e.g., *Circelli, Clark & Cronin-Golomb, 2013*; *Silver & Bilker, 2015*), while others distinguished the terms and used specific tasks to assess emotion identification and emotion recognition separately (*Benito et al., 2013*; *Mathersul et al., 2009*; *Wilhelm et al., 2014*). It is essential to use these concepts uniformly in future studies. In this meta-analysis, we applied the term emotion identification as the "ability to visually analyze the configuration of facial muscle orientations and movements in order to identify the emotion to which a

particular expression is most similar'' (*Wilhelm et al., 2014*). We assume that the term emotion recognition emphasizes a focus on memory for facial expressions of emotion, i.e., the ''ability to correctly encode, store, and retrieve information regarding emotional expressions from memory systems'' (*Wilhelm et al., 2014*). The ambiguity in the use of these terms may lead to misunderstandings during the phase of literature search and in the interpretation of the published results. In this sense, future studies should pay more attention to this issue.

## CONCLUSIONS

In sum, the present meta-analysis shows evidence of less accuracy of older adults in emotion identification, not supporting a positivity bias nor a reduction in the negativity effect. Meta-regression analyses suggest that effect sizes are moderated by sample characteristics such as sex, level of education, as well as stimulus features. Several factors might explain the age-related differences in emotion identification, but future studies are needed to explore whether and to what extent they are involved.

### Funding
This research was supported by a grant from the Fundação BIAL. Carina Fernandes was supported by a doctoral grant from the Fundação para a Ciência e Tecnologia (Carina Fernandes - SFRH/BD/112101/2015). The funders had no role in study design, data collection and analysis, decision to publish, or preparation of the manuscript.

### Grant Disclosures
The following grant information was disclosed by the authors:
Fundação BIAL.
Fundação para a Ciência e Tecnologia: SFRH/BD/112101/2015.

### Competing Interests
The authors declare there are no competing interests.

### Author Contributions
- Ana R. Gonçalves conceived and designed the experiments, performed the experiments, analyzed the data, prepared figures and/or tables, authored or reviewed drafts of the paper, approved the final draft, data extraction.
- Carina Fernandes conceived and designed the experiments, performed the experiments, prepared figures and/or tables, authored or reviewed drafts of the paper, approved the final draft, data extraction.
- Rita Pasion performed the experiments, prepared figures and/or tables, authored or reviewed drafts of the paper, approved the final draft.
- Fernando Ferreira-Santos and Fernando Barbosa conceived and designed the experiments, contributed reagents/materials/analysis tools, authored or reviewed drafts of the paper, approved the final draft.

- João Marques-Teixeira conceived and designed the experiments, analyzed the data, contributed reagents/materials/analysis tools, authored or reviewed drafts of the paper, approved the final draft.

## Data Availability

The research in this article did not generate any data or code— we performed a meta-analysis with data from previous studies.

## Supplemental Information

Supplemental information for this article can be found online at http://dx.doi.org/10.7717/peerj.5278#supplemental-information.

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
