# Peer review of "Effects of age on the identification of emotions in facial expressions: a meta-analysis"

_PeerJ, doi:10.7717/peerj.5278_

## Round 0.1 · original submission · Major Revisions

As you can see, both reviewers have a number of major concerns with the manuscript in its current form that will require substantial revisions to address. If you are able to address these concerns then please be sure to respond to each point in a detailed rebuttal / revision.

Reviewer 1 ·

Basic reporting

BASIC REPORTING
Introduction
• Lines 60-62: This statement doesn't really reflect the literature on aging and emotion processing. There are many studies showing the effects of age on emotion processing, for example, the well-known positivity effect among older adults has been established in several studies (e.g. Williams et al., 2006). In addition, there are studies that assess how cognitive processing in old age influences emotion identification (see Mitchell, Kingston, & Barbosa Bouças, 2011; Orgeta, 2010; Orgeta & Phillips, 2008; Sullivan & Ruffman, 2004 for example). It would be beneficial for authors to delineate exactly what they mean by both emotion identification and emotional processing at the beginning of the manuscript, and whether or not they are treating these terms separately throughout the paper.
• Lines 74 onwards: This part of the introduction discusses age-related positivity biases in relation to memory. It is unclear how this links into the question of how old age influences emotion identification, which is addressed in the current paper. The authors should either omit the details on age differences in emotion memory or make a direct link between the literature on positivity effects in memory and emotion perception in old age. Later (lines 85-89) there is also mention of studies on age and the dot probe task – it is not made clear what construct these studies measure or how it relates to emotion identification.
• Lines 90-101: Again, it would be useful for the authors to outline exactly what they mean by “emotional processing”, and whether or not they are using “emotion identification” and “emotion processing” interchangeably.
• Lines 101-103: The authors highlight the need to study age differences in dynamic vs. static stimuli, and this is an interesting point to pursue further. However, this seems to be the only point where this issue is discussed. The authors should consider discussing whether this has an impact on age effects in emotion identification in more detail in the introduction.
Overall the introduction should review in more depth the previous literature on emotion identification, linking more clearly into the aims of the current study. Literature on emotion memory and dot-probe should only be discussed if it can be shown to be relevant to the aims. It would be nice in setting out the literature and aims to distinguish between possible participant characteristics influences and task influences on emotion identification.

Experimental design

EXPERIMENTAL DESIGN
Method
• Generally, exclusion and inclusion criteria are clearly reported.
• The PRISMA diagram is a useful addition to the method section.
• The statistical analysis section should be presented in full paragraphs.

Validity of the findings

Results
• What was the overall effect size for age across all emotions?
• The stimulus moderating effects described in the analysis section need to be presented in tables and text. This will add real strength to the paper, whether or not the effects are significant.
• The text describing the meta-regression statement is unclear and not detailed enough. Do the results mean that females were better at identifying fear than males but worse at identifying disgust? It would be good for the authors to clarify. It would be useful to be more precise when talking about ‘effects’.
Discussion
Much of the discussion concentrates on speculation about age differences in emotion identification unrelated to the results of the current study. It would be more useful to outline and discuss the key findings from the meta-analysis at the beginning of the discussion before going on to speculate about possible links to the brain. I missed proper discussion and interpretation of the key effects of gender and stimulus types.
• Lines 188-190: It is not clear that the findings disagree with those of Ruffman et al. where an age difference in happiness identification was also reported.
• Lines 191-196: The authors compare findings from emotion identification tasks to the literature on age-related positivity effects. It would be useful to further suggest how the tasks used to assess emotion identification are different from tasks used typically to assess positivity effects (for example, many studies assess positivity effects using memory paradigms, see Mather & Carstensen, 2005) - not many studies have addressed the issue of whether typical emotion identification labelling tasks tap into age-related positivity effects and it would be good if the authors could consider how this task may be insensitive to positivity effects.
• Lines 203-207: The authors suggest that their results are consistent with prior findings on reduced amygdala activation associated with memory for negative stimuli, however findings suggested that older adults are typically worse at identifying a handful of negative expressions and happiness. It is unclear how a decline in the sensitivity to memorise negative stimuli influences identification of happiness. It would be good if the authors could highlight why this process may not be affecting older adults' perception of happiness. Are there perhaps separate causal mechanisms underlying identification of happy expressions?
• Lines 215-219: The authors highlight that a reduction in the tendency to look at the mouth region of faces might cause difficulties in recognising facial expressions for older adults, but how does this explain the age-related differences in recognising emotions that are predominantly expressed via the mouth region, such as happiness, sadness?
• Lines 225-228: This is an interesting observation, and it would be good if the authors could expand on this point - why might differences in education be important for emotion identification? Expanding on this point further may help the authors highlight the novelty of their study.
• Lines 244-251: While I agree that it is essential to use terms uniformly, aging studies more often adopt the term "emotion recognition" rather than "emotion identification" to describe the ability to identify emotions (see for example Franklin & Zebrowitz, 2017; Grainger, Henry, Phillips, Vanman, & Allen, 2017; Richter, Dietzel, & Kunzmann, 2010; Ruffman, Murray, Halberstadt, & Taumoepeau, 2010; Sullivan & Ruffman, 2004; Sze, Goodkind, Gyurak, & Levenson, 2012; Wieck & Kunzmann, 2017). It therefore seems to make more sense to adopt this terminology to ensure consistency with the literature.
Conclusion
• Lines 258-259: The finding that females were better at recognising some emotions than others was not adequately discussed in the discussion section. It would be good if the authors could address this issue further, and suggest how this influences age-related differences in emotion identification.

Additional comments

Summary
This paper reports a meta-analysis of 24 studies examining the effects of age on the ability to identify facial expressions of emotion. Findings suggested that older adults are worse than younger adults at identifying anger, fear, sadness, surprise and happiness. Furthermore, education moderated the effects of age on emotion identification.
The authors should be commended for recognising the need to conduct an up-to-date meta-analysis on the effects of age in facial emotion identification. As such, the findings provide a useful extension of the findings reported by Ruffman, Henry, Livingstone and Phillips's (2008) meta-analysis. The conclusion that sample characteristics are important is novel, and should be expanded upon. Overall, we recommend that the authors make some amendments to the manuscript (see below) to improve clarity and impact of the paper. Comments have also been made to the pdf (see attached).

References from review:
Franklin, R. G., & Zebrowitz, L. A. (2017). Age Differences In Emotion Recognition: Task Demands Or Perceptual Dedifferentiation? Experimental Aging Research, 43(5), 453–466. https://doi.org/10.1080/0361073X.2017.1369628
Grainger, S. A., Henry, J. D., Phillips, L. H., Vanman, E. J., & Allen, R. (2017). Age Deficits in Facial Affect Recognition: The Influence of Dynamic Cues. Journals of Gerontology - Series B Psychological Sciences and Social Sciences, 72(4), 622–632. https://doi.org/10.1093/geronb/gbv100
Mather, M., & Carstensen, L. L. (2005). Aging and motivated cognition: The positivity effect in attention and memory. Trends in Cognitive Sciences, 9(10), 496–502. https://doi.org/10.1016/j.tics.2005.08.005
Mitchell, R. L. C., Kingston, R. A., & Barbosa Bouças, S. L. (2011). The specificity of age-related decline in interpretation of emotion cues from prosody. Psychology and Aging, 26(2), 406–414. https://doi.org/10.1037/a0021861
Orgeta, V. (2010). Effects of age and task difficulty on recognition of facial affect. The Journals of Gerontology: Psychological Sciences, 65B(3), 323–327. https://doi.org/10.1093/geronb/gbq007.Advance
Orgeta, V., & Phillips, L. H. (2008). Effects of Age and Emotional Intensity on the Recognition of Facial Emotion. Experimental Aging Research, 34(1), 63–79. https://doi.org/10.1080/03610730701762047
Richter, D., Dietzel, C., & Kunzmann, U. (2010). Age differences in emotion recognition: The task matters. Journal of Gerontology: Psychological Sciences, 66(1), 48–55. https://doi.org/10.1093/geronb/gbq068.
Ruffman, T., Henry, J. D., Livingstone, V., & Phillips, L. H. (2008). A meta-analytic review of emotion recognition and aging: Implications for neuropsychological models of aging. Neuroscience and Biobehavioral Reviews, 32(4), 863–881. https://doi.org/10.1016/j.neubiorev.2008.01.001
Ruffman, T., Murray, J., Halberstadt, J., & Taumoepeau, M. (2010). Verbosity and emotion recognition in older adults. Psychology and Aging, 25(2), 492–497. https://doi.org/10.1037/a0018247
Sullivan, S., & Ruffman, T. (2004). Emotion recognition deficits in the elderly. International Journal of Neuroscience, 114(3), 403–432.
Sze, J. A., Goodkind, M. S., Gyurak, A., & Levenson, R. W. (2012). Aging and Emotion Recognition : Not Just a Losing Matter, 27(4), 940–950. https://doi.org/10.1037/a0029367.Aging
Wieck, C., & Kunzmann, U. (2017). Age differences in emotion recognition: A question of modality? Psychology and Aging, 32(5).
Williams, L. M., Brown, K. J., Palmer, D., Liddell, B. J., Kemp, A. H., Olivieri, G., … Gordon, E. (2006). The Mellow Years?: Neural Basis of Improving Emotional Stability over Age. The Journal of Neuroscience, 26(24), 6422–6430. https://doi.org/10.1523/JNEUROSCI.0022-06.2006

Annotated reviews are not available for download in order to protect the identity of reviewers who chose to remain anonymous.

Reviewer 2 ·

Basic reporting

I commend the authors on conducting a meta-analysis. However, the main concern I have is that only data published since 2008 are analysed, thus neglecting the body of work prior to that date. This provides a potentially misleading account of the overall effect of age on the identification of facial expressions of emotion. There is also no reference to any attempt to identify and integrate unpublished research. This leads to the ‘file drawer’ problem and a potential over-representation of confirmatory results. Furthermore, if the current study is examining differing moderators to the previous Ruffman et al. meta then it would make sense to include the studies that were included in Ruffman’s paper.

The Introduction suggests that Ruffman et al. found an age-related decline in recognition of all expressions except happiness (line 81). I don’t think this is correct. Ruffman et al. state “older adults were worse on fear, surprise and happiness in some modalities… In contrast, recognition of disgust seems relatively preserved” (p. 971). This is also reflected in Ruffman’s Table 1 particularly in relation to recognition of emotion just in faces.

The influence of cognitive (attention and memory) and sensory (visual acuity) processes on emotion identification are referred to in the Introduction. However, neither of these variables are assessed in the meta-analysis. It would be preferable to focus the Introduction on the variables that are examined. These appear to be briefly referred to in the last paragraph of the discussion and instead should be elaborated on earlier throughout the Introduction.

Also in the final paragraph of the Introduction, when referring to previous effects of college education, for example, please clarify whether those effects were found in young and/or older adults. Then, also please describe the predicted effect of each of those moderators on age differences in emotion recognition.

Experimental design

The paper explains that it fills a gap in the literature by (1) conducting the meta on studies published since the previous meta on this topic and (2) examining different moderators to the previous meta. But, I believe all studies up to the search date should be analysed to fulfil this aim. In addition, the moderators being tested seemed to change throughout the paper and were not sufficiently introduced in the Introduction or described in the Results or Discussion. It is therefore difficult to see how this paper does fill any gaps.

Why were only studies that allowed effect size data to be directly recorded, calculated or measured included (criterion 2; line 122)? Why were authors not contacted for relevant data when it was missing from a manuscript, as is standard practice? [I now see this mentioned on line 140, so perhaps line 122 needs further clarification?]

Please specify how many articles for which researchers read the full text (line 126), and does full text mean the full article?

Was a search of studies citing the Ruffman meta conducted?

There needs to be an explanation of what a positive (versus negative) effect size represents.

Validity of the findings

Table 4 indicates that sex and education were entered as potential moderators. Were the other potential moderators outlined at the end of the introduction also examined? The other moderators are also not mentioned in the Results text.

In the Discussion, moderators should be discussed earlier, and before reference to potential neural mechanisms which were not the focus of the current meta.

Line 215 refers to differences between young and older adults with respect to how they examine stimuli and their physiological responses. But neither of these things were measured in the current meta. Please clarify further or remove this section.

Line 222 the significant association with sex is not elaborated.

I’m not sure how the paragraph concerning emotion recognition versus identification fits with the aim of the present meta.

Additional comments

It is stated on line 72 that “we will discuss below, an obvious memory improvement for positive faces”. But the following paragraph refers mainly to emotion recognition rather than memory.

On line 81, what is meant by “posterior studies”?

The paragraph starting on line 85 would be better integrated with the earlier paragraph about the positivity effect.

The final sentence of the Introduction is unclear.

---

## Round 0.2 · accepted · Accept

Thank you for carefully addressing the reviewers concerns. Reviewer 1 was happy with the changes made, as was I, and so I considered Reviewer 2’s feedback in light of that and decided that the revised paper was acceptable.

# Reviewer 1 ·

Basic reporting

This new version is much improved.

Experimental design

The design is appropriate.

Validity of the findings

The interpretation of the results is much clearer now.

Additional comments

The authors have dealt well with the reviews.

Reviewer 2 ·

Basic reporting

NA

Experimental design

NA

Validity of the findings

NA

Additional comments

My main concern from the initial paper was not sufficiently addressed. That is, the data included in this meta-analysis only dates back to 2008. This excludes a large body of the existing data, thus potentially providing a misleading account of the effects being tested. I believe this can only be rectified by adding all of the excluded data into the meta-analysis. Alternatively, the paper could be presented as a systematic review (without effect sizes) that refers back to previous reviews/metas.